# Finding the Proportion of Females with Autistic Spectrum Disorder Who Develop Anorexia Nervosa, the True Prevalence of Female ASD and Its Clinical Significance

**DOI:** 10.3390/children10020272

**Published:** 2023-01-31

**Authors:** Robert McCrossin

**Affiliations:** 1Cooroy Family Practice, Cooroy 4563, Australia; bobmccrossin@gmail.com; 2Beerwah Family Clinic, Beerwah 4519, Australia

**Keywords:** autistic spectrum disorder, anorexia nervosa, female, prevalence, Bayes’ theorem, likelihood ratio, hazard ratio, diagnosis, management, generalized joint hypermobility

## Abstract

It appears that up to 80% of females with autistic spectrum disorder (ASD) have not been diagnosed by the age of 18. This translates to a prevalence of about 5–6%, and if true, has serious implications for female mental health. One way of finding the true value is to use Bayes’ Theorem with a comorbid condition as a more easily recognizable flag. An obvious choice is anorexia nervosa (AN), but it transpires that the proportion of women with ASD who develop AN is unknown. This study uses published data in novel ways to provide two methods of estimating a range for this variable, and gives a median value of 8.3% for AN in ASD and, with four other methods, a median prevalence of 6% for female ASD. The clinical implications of the diagnosis and management of ASD and its comorbidities are discussed and, as an example, a solution is provided for the rate of ASD in symptomatic generalized joint hypermobility. It is probable that one in six women with a mental health condition is autistic.

## 1. Introduction

### 1.1. Preamble

Autistic spectrum disorder (ASD) is a life-long neurodevelopmental condition with problems in socialization and communication, restrictive interests and repetitive behaviors. It is highly genetically determined and underdiagnosed in females with negative long-term consequences [1]. In a previous paper [1], identifying biases against the recognition and diagnosis of ASD females, the estimated female population prevalence was about 6%, much higher than current estimates. This result was cross-checked by identifying more visible comorbidities (borderline personality disorder, BPD) and using BPD as a test for the prevalence of female ASD, employing Bayes’ theorem (Section 1.3). The value found was 6.0%. This paper will similarly explore this, using anorexia nervosa (AN) in the same way. AN is a serious eating disorder, predominantly in females, which is life-threatening if neglected [2]. Compared to female ASD it is relatively easy to detect (see Section 2.5). The value of finding the true prevalence of ASD is that it may be a common comorbidity of many mental health conditions and not recognizing ASD generally makes clinical management more difficult and less successful.

The aim of the study is to explore the use of AN epidemiological data with Bayes’ theorem to more firmly establish the true prevalence of female ASD and thereby improve the clinical management of these complex comorbidities. To this end, it will include a discussion on the current aetiology and structure of mental health conditions and clinical approaches to the problems of ASD management.

An essential intermediate step examines data detailed in current peer-reviewed literature, to find solutions to the proportion of females with ASD who develop AN at some time in their lives, designated P(AN|ASD). P stands for probability, in this case, a conditional probability, but it is equivalent to prevalence in the population of interest. Proportion, in context, may be a synonym for both. P(AN|ASD) would provide further information on the true prevalence of ASD in females, designated P(ASD). The study provides new epidemiological information from the patient dataset detailed previously [1]. It looks at published information in novel ways consistent with Bayesian reasoning and is, therefore, an exercise in quantitative induction [1]. The task faces four major challenges:

Most literature on ASD epidemiology does not report gender-specific data in sufficient detail. ASD is very diverse and gender-related issues require gender-specific approaches. Gender-nonspecific data in ASD is generally as useful as the observation that the average human has one ovary and one testis. You cannot average different entities, and for virtually all practical purposes, female and male ASD are sufficiently different, so averaging is meaningless. An underlying criterion, such as poor reciprocal communication, will be shared but the pattern of responses will be different in quality and quantity. The gender data should be reported separately and combined in the few instances where this is applicable.There is virtually no published data on P(AN|ASD), and none clearly specific to females. P(AN|ASD) could be an important intermediate step in clarifying P(ASD) because AN in females is much easier to recognize as a problem than in female ASD and can act as a flag for finding P(ASD). This will be a major theme of this paper.Since female ASD is cryptic, it is likely that AN will be diagnosed first in a clinic dedicated to its usually urgent treatment. Whether ASD is diagnosed prior to or subsequent to AN, it is likely to be managed by the mental health arm of that clinic. There is not going to be the denominator of a representative ASD population to provide the overall proportion of AN cases. It is not then surprising that there is virtually no published value for P(AN|ASD).The only published quantitative estimates of bias in ascertaining P(ASD) are in the recent paper [1] and there need to be independent bias-free estimates of P(ASD) to firm up an agreed value to assess its clinical importance, which is likely to be high. Finding methods to do this is an important task of this paper.

### 1.2. Epidemiology of Female ASD

There is a lot of confusion in reporting epidemiological information as outlined in challenge (1) above. This section will hopefully clarify the issues and provide context for the study. It will focus on female ASD and provide data from my patient database of 1711 children 1–18 years of age outlined in reference [1]. This reference includes details of my pediatric practice. It will examine data over time and in two groups. The first major group is termed singletons and includes two subsets consisting of (1) the first female diagnosed in a family (proband) and (2) the only female diagnosed (simplex). The assumption is that, in terms of recognition, they are equivalent. The total singletons will then be divided into two subgroups, the first referred for diagnosis specifically because ASD has been considered (designated ?ASD). The second subgroup is referred to because there is a learning problem, an undifferentiated behavior problem, or another diagnosis where ASD is found to be present, commonly one of the varieties of attention deficit disorder which has a very high overlap with ASD (designated behavior/learning).

The second major group is siblings. These are all children with the same carer who are diagnosed after the proband. It will include half-siblings and occasionally step-siblings or foster siblings because the issue is recognition, not genetic relatedness. The assumption is they are easier to recognize than singletons due to family experience of ASD.

A small number of children did not have a settled core gender identity and were classified by their current sex of assignation. Only those assigned as female are counted in this study. 

The first problem is that overall prevalence is generally reported as either up to 8 years or over the entire child and adolescent period, and while the value up to age 8 may be useful for subgroup comparisons, Figure 1 shows it does not give a useful estimate of overall prevalence even within the pediatric age group. The autistic condition is permanent and not fatal so the prevalence of ASD will continue to increase in a population cohort as long as new diagnoses are being made. The cumulative incidence measured at age 8 is still rising steadily [3]. The only caveats would be premature death or therapeutic transformation of ASD to a non-disordered autistic spectrum condition (ASC) [1]. The graphs will show time-related patterns and Table 1 will list the median age of diagnosis for up to 96 months and up to 18 years, and the relative proportions of each age group in each diagnostic group.

The total singleton graph (Figure 1) is fairly smooth though there appears to be a possible double peak at 5 years and 10 years. This accords with clinical experience. The onset of school and the onset of puberty are stressful times for girls and the fact there is a problem becomes evident. There are, however, two different subgroups (Figure 2). The first is where ASD is suspected and is the reason for referral. The pattern is clearly biphasic. A subset at ~3–5 years either has very clear features of ASD or there is some knowledge of ASD within the family that triggers a referral (?ASD). If they are missed, then camouflaging at school is quite effective [1] until they enter puberty where the extra stress makes the problem evident. Those in the other subgroup are those picked up at school with either behavior or learning problems and correctly diagnosed as ASD by the clinician (behaviour/learning). This group is often initially diagnosed as attention deficit disorder, which may be present but acts as a distractor for the comorbid ASD, in particular for girls where the diagnosis may be delayed for over 2 years [4]. This may be further complicated because attentional problems are probably a feature of ASD [5]. There appears to be a possible extra pubertal peak at around 13 years which makes sense since they are not likely to be picked up as early as the ?ASD pubertal subset, which shows clearer features of ASD.

The sibling group is compared to the singleton group in Figure 3, plotted 6 monthly to show finer detail. The most striking difference is the number of siblings who are diagnosed very early. This is due to the fact that the carers have already gone up the learning curve and know what to look for. This was the basis for my estimate of the true male–female odds ratio (MFOR) [1]. There is a small peak at 17 years which is a function of getting a last-minute pediatric assessment where the family may have been prevaricating, in particular with an older sister, since it is much easier to get a pediatric than an adult diagnosis in Australia.

Table 1 shows the comparison of time to diagnosis in each group. It also clearly shows that the sibling group is diagnosed earlier than the singletons.

The difference in proportions of singletons diagnosed by 96 months of 39.7% and siblings diagnosed of 48.6% was significant by χ^2^ for proportions. χ^2^ = 5.087 DF = 1 φ*_c_* 0.089 *p* = 0.024.

### 1.3. Bayes’ Theorem

While the proportion P(AN|ASD) of females with autistic spectrum disorder who develop anorexia nervosa at some time in their lives is not known, the reverse conditional probability P(ASD|AN) has recently received a lot of attention and current estimates vary from about 20–30% [2,6,7]. This has been an important development because the underlying etiology of anorexia is probably different, with important implications for therapy (Section 3.2). AN is a very visible clinical condition which is hard to miss, whereas ASD in females is a cryptic condition which is very easy to miss [8,9]. The majority of diagnoses of ASD in females are made in childhood and adolescence with a median age in my clinic, for example, of 8 years 4 months (range 1–18). Adult ASD services are often fragmented [10] and pediatric handover is inconsistent. The peak of anorexia nervosa diagnosis in females is around 15–19 years [11]. The combination of uncoordinated services for adult ASD in particular and asymmetric temporal recognition of the two conditions makes it very hard to assemble a representative ASD population to study. Once AN is diagnosed however it is relatively easy to assess the proportion P(ASD|AN) who have ASD. 

Bayes’ theorem gives a general method of determining the prevalence of conditions that may be difficult to detect clinically. If there is a more easily identifiable condition associated with the cryptic one, three components of the simplest form of the theorem have the visible condition within them. In the case of cryptic ASD these are:

The prevalence of the visible condition P(AN). Prevalence estimates should be reliable.The proportion of the visible condition AN in the population of the cryptic condition ASD P(AN|ASD). Despite AN being visible, P(AN|ASD) is unknown as per challenge 3. Section 2.2 will show how this difficulty may be overcome.The proportion of the cryptic condition ASD in the population of the visible condition AN: P(ASD|AN). Because AN is visible, ASD can be carefully sought and therefore, P(ASD|AN) is reliable.

Bayes’ theorem connects the two conditional probabilities P(AN|ASD) and P(ASD|AN) and the respective prevalences of the two conditions P(ASD) and P(AN). The visible condition acts as an identifiable flag to determine the prevalence of the cryptic condition with Bayes’ theorem:P(AN|ASD) × P(ASD) = P(ASD|AN) × P(AN)

Definitions of mental health conditions can be tricky to interpret but as long as the interpretations are reasonably consistent in each variable, P(ASD) will be reasonably accurate. The assessment time frame is very important if the diagnostic timing is different, as is the case here, and so, results are going to be more accurate with data, at least into the third decade to give time for AN to evolve and be found. 

This study will use three forms of Bayes’ theorem with two different though overlapping datasets to find P(AN|ASD). The first form is based on the simple algebraic relationship of probabilities used as a guide in clinical medicine. This first version provides an exact result for the unknown variable P(AN|ASD) if the other three are known. While there may be multiple underlying estimates of each independent variable, each needs to be reduced to a single number for each individual calculation. Plausible ranges will be dealt with in Section 2.1 method one.
P(AN|ASD) × P(ASD) = P(ASD|AN) × P(AN)

What the theorem does is look at the intersection of the conditions, patients who have both ASD and AN, and assign each element of this pair as a proportion of the other underlying condition. These conditional probabilities depend on the relative proportions of each condition in the population, i.e., the prevalences of AN and ASD. The condition with the greater prevalence has a lower comorbid proportion of the other condition. If as shown in Figure 4, P(ASD) > P(AN) then P(AN|ASD) < P(ASD|AN) in inverse proportion.

The study will use data for lifetime prevalences, as far as these are known, but the extent of both AN and ASD over the lifetime is still not well characterized, and in practice, this does not extend much beyond the third decade.

Since AN is far easier to recognize than ASD in females, the clinical value of P(ASD|AN) is high. The proportion of ASD is clinically significant (Table 2) and there is evidence that these patients do poorly unless the underlying ASD is recognized [12]. We will use this simple algebraic form of Bayes’ theorem to estimate P(AN|ASD) in Section 2.1.

The second form of Bayes’ theorem employed is the 2 × 2 table (Figure 5), commonly used to assess the outcomes of a clinical test for a particular condition. This provides a probability that a positive test reflects the presence of a condition or a negative test reflects its absence. These are the positive and negative predictive values, respectively. 

The 2 × 2 table also provides likelihood ratios. The positive likelihood ratio (LR+) is the ratio of the probability that the positive test result is a true positive for the condition present and the probability that the positive test result is a false positive for the condition absent, given by: (a/(a + c))/(b/(b + d))
or:sensitivity/(1 − specificity)

The positive likelihood ratio, also known as the Bayes’ factor, will be employed in a novel way in Section 2.2 method two in a 2 × 2 table to provide possible exact solutions for P(AN|ASD) and P(ASD). This result will be confirmed with the third, or odd forms of Bayes’ theorem (see Section 2.2).
Posterior odds = likelihood ratio × prior odds.

## 2. Methods of Assessing P(AN|ASD), P(ASD) and Results

### 2.1. Method One

When searching the literature for the required values of the independent variables, we find they are very variable indeed. There may be some variation between the study populations but there are also wide variations within populations that are likely fairly homogeneous. The problem is that there are variations both in the criteria for diagnoses, the interpretation of those criteria for a given diagnosis (a particular problem for high-functioning female ASD) and in the age ranges which are assessed. The aim of method one is to model this reality and see where it leads. The three independent variables needed to establish the dependent variable P(AN|ASD) are P(ASD|AN), P(AN) and P(ASD). Information was sought in the literature to establish reasonable ranges for these variables which would then be put into a matrix representing the variation in estimates that would accrue if direct clinical estimates of P(AN|ASD) were ever done. If these ranges are reasonably accurate this would model a very large number of potential studies with widely varying results, and the median would likely not be too far off a consensus value that might take many years to establish by direct observational studies. My selected ranges were, of course, educated guesses but were based on the current available data for AN lifetime prevalence [13], P(ASD|AN) [2,6,7] and prevalence of ASD. For the first two, there are quite a lot of studies and the ranges can be established with reasonable confidence. This is not so for ASD prevalence and the range is based on assessments from the previous study of the proportion of females missed [1], where a range of possible P(ASD) prevalences can be calculated both from my data and independent data as explained in that study.

The estimated ranges for the Bayesian analysis are listed in Table 2.

These values are not weighted because with the wide variation, there is no reason to suppose any value is more likely than any other and the median should weigh the results. There is also no reason to suppose that the results will be normally distributed.

The matrix of 27 results for P(AN|ASD) is shown in Table 3 with the median result of 0.091 in red. 

The median value (in red) is not in the centre of the numerical range and so the values were then plotted as percentages on a histogram to establish the pattern of results (Figure 6).

The median is skewed to the lower end of the range. Plotting intermediate independent variables for a total of 125 estimates smoothed the peaks and reinforced the skew to the left.

It appears about 9% of females with ASD will develop AN at some stage. The weak link in this assessment, of course, is the absence of bias-free assessments of P(ASD) independent of my study. There are two approaches to this problem. The first is to see what P(AN|ASD) values would arise compared to the value of 0.091 using current literature estimates of P(ASD) and decide whether they are plausible. The second is to seek any literature at all that can establish conditional probabilities in both directions and might produce another estimate of P(ASD).

A recently published estimate of P(ASD) is 0.0125 [14]. Used with the values of P(ASD|AN) and P(AN) which derived P(AN|ASD) of 0.091 (9.1%) this would give P(AN|ASD) as 40%. The 2018 Australian Bureau of Statistics value for P(ASD) [15] of 0.004 (0.4%) would give a value of 125% which, of course, is impossible. For this prevalence, even the lowest estimates of P(ASD|AN) and P(AN) would give a value of P(AN|ASD) of 50%. One would think proportions of this magnitude would have been detected in some clinics despite the logistic difficulties.

### 2.2. Method Two

A Danish study in 2015 [16] looked at aspects of the relationship between ASD and AN. It was a nationwide register-based cohort study and thus had very large numbers of widely distributed subjects. It did not estimate conditional probabilities directly but it did look at the relationship between ASD and AN from both directions. Inter alia the study reported a hazard ratio of 5.22 in females comparing the rate of AN in the presence of ASD and in the absence of ASD. This method will show how both P(AN|ASD) and P(ASD) can be derived using the hazard ratio with Bayes’ theorem. The range of ages in this study is from birth to 27 years. There is no information to determine probabilities beyond this age which is a general caveat for “life time” values.

The registers were the Danish Psychiatric Central Research Centre (from 1969) and the Danish National Patient Register (from 1977). The classification systems were the International Classification of Diseases (ICD-8) until the end of 1993 and then ICD-10. The study period was from 1981 until 2008. The study population was large, with 4684 females with AN and 2594 females with ASD. The ASD females were probands, i.e., the first child in the family diagnosed. AN included atypical anorexia and ASD included infantile autism, atypical autism and Asperger’s syndrome. These latter three would likely be now subsumed into ASD as defined in the Diagnostic and Statistical Manual version 5 (DSM-5) [17]. Since both groups in the hazard ratio have AN the ratio including atypical AN would not change unless P(AN|ASD) is different from P(atypical AN|ASD). There is evidence that the difference is not significant [18]. 

Method two will give a measure of the proportion of AN diagnosed as representative of a population of ASD as possible since it was effectively a census of the entire Danish population. This then overcomes the problem of finding P(AN|ASD) in a representative ASD population. It assumes that the rate of AN in the diagnosed population of ASD is representative of the rate in the entire ASD population. This is probably largely true, since the major reason for the failure to ascertain female ASD appears to be non-recognition of the possibility of ASD prior to referral, rather than failure of the diagnostic process itself [1]. 

A hazard ratio looks at how the rates of occurrence of the two conditions compare over time. The study adjusted for multiple variables in eight consecutive cohorts to arrive at a final single numerical hazard ratio of 5.22 for individuals with ASD getting a second diagnosis of AN, compared to individuals without ASD getting a diagnosis of AN. Having allowed for the variability in the hazard ratio with large numbers over time, this would be similar to a relative risk. If the estimated range of prevalences of AN is 1–3% this would give a range of P(AN|ASD) of 5.22–15.66%. In Figure 7, we compare the range of values of methods one and two.

There appears to be a very symmetrical overlap of the ranges. Is it possible to establish a comparable value in method two to the matrix median in method one?

The hazard ratio was defined in the study as the ratio of getting a diagnosis of AN with ASD compared to getting a diagnosis of AN without ASD. This is:P(AN|ASD)/P(AN|not ASD)

With a 2 × 2 table, the ratio of getting a true positive result with a test compared to getting a false positive result with the test is the Bayes’ factor or positive likelihood ratio. The general 2 × 2 table, including the positive likelihood ratio, is shown in Figure 8 and the table for method one is shown in Figure 9.

If we say AN is a visible flag for cryptic ASD, this is analogous to saying the presence of AN is a test for the presence of ASD and we can construct a 2 × 2 table to examine the relationship. We use the second condition (AN) as a test for the first condition (ASD). It is the mathematical equivalent of any other test. The sensitivity of AN as a test for ASD is mathematically P(AN|ASD). It is the probability of getting a true positive result when performing the test which translates to the probability of developing AN if you have ASD. The false positive rate of the test is P(AN|not ASD). The likelihood ratio is:P(AN|ASD)/P(AN|not ASD)

Mathematically, the hazard ratio and the positive likelihood ratio are equivalent. Providing the single value of the hazard ratio is a valid measure over the total data-collection period and the relation of ascertainment of ASD and AN has not changed significantly since the data were collected, we can use the hazard ratio as a likelihood ratio with the current estimates of the independent variables. There has probably been little change in female diagnostic bias during this time [1]. We will now explore the effect of the hazard ratio as a positive likelihood ratio.

P(ASD|AN) is, mathematically, the positive predictive value and in its odds form is the posterior odds used with the Bayes’ factor (LR+) to find the prior odds and thus the ASD prevalence P(ASD). 

The positive likelihood ratio for the matrix method relationship is 5.73. This is quite similar to the hazard ratio and implies that the relative recognition of ASD and AN has been similar over time. Then, with a “likelihood ratio” of 5.22 our new result for P(AN|ASD) should be a little lower than 0.091. As can be seen, the “sensitivity” of the test is very low, but the “specificity” is so high (0.984) that the “likelihood ratio” is a significant number. The constant independent variables for each method are the current best-established mid-point estimates in the literature, i.e., the prevalence P(AN) of 0.02 and P(ASD|AN) of 0.25. To find the new P(AN|ASD) we need to adjust the prevalence of ASD. This will give a new estimate of this variable. The derivation is shown in Appendix A. The 2 × 2 table for method two is shown in Figure 10.

The value for P(ASD) can be confirmed using the odds form of Bayes’ theorem:Posterior odds = likelihood ratio × prior odds

As discussed, we use the equivalence of the likelihood and hazard ratios and the equivalence of probability and prevalence. The conversion equations for probability P and odds O are:O = P/(1 − P) and P = O/(1 + O)

Prior odds of ASD in the general population = posterior odds of ASD in the AN population/hazard ratio:
Odds(ASD) = (0.25/(1 − 0.25))/5.22         = 1/(3 × 5.22) = 0.0639
P(ASD) = 0.0639/1.0639    = 0.060

The two ranges for P(AN|ASD), the method one median and hazard ratio-derived values are shown in Figure 11.

### 2.3. Derivation of Overall Median Values for P(AN|ASD) and P(ASD)

An important result of this study was the two new derivations of the prevalence of female ASD. The results for P(ASD) from the previous study [1] were used in method one, but the close agreement of P(AN|ASD) in method two using new information provides support for the previous findings. The main aim of the previous study was to estimate the biases preventing the diagnosis of females and find the proportion of girls missed, and the latter result was obtained by two general methods. 

In the first method, values from the previous study using my own data were derived from the application of the total bias against diagnosing girls in two groups, first diagnosed and only diagnosed. This would be similar to the Danish group in method two of this study. Another estimate by the same method uses the fact that siblings are easier to recognize than the first child diagnosed in a family, due to parental experience, and this means fewer siblings are missed and reduces the overall bias. The calculation of the revised proportion missed is shown in Appendix B and enables another estimate of P(ASD) using the whole female ASD population. There is then a lower prevalence estimate as shown in Appendix B.

The second method compares values for girls missed which were found by other methods against the latest ASD prevalence value published at the time [14] of 1.25%. The calculation is also shown in Appendix B. 

This study and the previous one [1] have given a total of 8 estimates for the true value of the prevalence of female ASD (Table 4). The matrix study value is included because although P(AN|ASD) was derived from the range of P(ASD), determined on the basis of the results of the previous study, new information on the values of P(ASD|AN) [2,6,7] and P(AN) [13] gave a value for P(AN|ASD) which was very clearly consonant with the other derived P(AN|ASD) values (0.091 with median value 0.084) (Table 4). The new information giving P(AN|ASD) means this reasoning was not circular because P(ASD) was “reverse engineered”. The estimates of the dependent variable P(ASD) all have intersecting sources of information but none are the same. All have new information to contribute to the overall median values of P(AN|ASD) and P(ASD). Two sets of values (Bayes’ and borderline personality disorder, and Bayes’ and hazard ratio) use data entirely independent from my own. There is no reason to favor any of the estimates and Table 4 shows the full list of solutions and medians of P(AN|ASD) and P(ASD) estimates. Figure 12 shows the different pathways to the estimation of P(ASD). 

Outline summary of the methods used to derive female P(ASD).

Find the biases in three different populations with ASD (probands, singletons and total female population [1]), calculate the proportion of girls missed and compare with the biased female prevalence [14] to calculate the range of possible true prevalences.Find the proportion of girls missed in methods 1,3 and 6 and compare the average biased male-to-female odds ratio (MFOR) calculated from these proportions [1] with the reported MFOR [14] to calculate the true prevalence.Calculate the true prevalence of borderline personality and ASD data with Bayes’ theorem [19,20,21].Use the hazard ratio as a likelihood ratio with the best published average estimates of P(ASD|AN) and P(AN) to calculate the true prevalence with Bayes’ theorem.Model clinical studies [10,11,12] to find a median estimate of P(AN|ASD) and reverse engineer Bayes’ theorem to estimate a plausible true prevalence.Calculate the true MFOR [1] and compare to the [14] male prevalence to calculate the true prevalence. This assumes all males are ascertained.

We can use the plausible range of P(ASD|AN) in its odds form together with the hazard ratio of 5.22 to estimate a plausible range of P(ASD) as shown in Table 5. 

On a different axis, we can use the 95% confidence interval for the hazard ratio of 4.11 to 6.64 [16] together with the midrange P(ASD|AN) of 0.25 to estimate another range of P(ASD) in Table 6.

These P(ASD) ranges are remarkably consistent with the range derived from the total of six methods in Table 4 (0.048–0.075), five of which do not use the hazard ratio as a likelihood ratio. 

### 2.4. New Information

#### 2.4.1. Updated Prevalence Data

During the manuscript preparation, new US data on increased ASD prevalence for 2019 and 2020 became available [22], from the same source as the data from 2014–2016 [14] used for calculations in Table 3. It may be that female recognition has improved due to the recent attention paid to female camouflage, though the male value increased even more. My database was from October 2014 to April 2020 so it overlapped with the new information. The calculations for P(ASD) were redone to see what effect there was with the new biased P(ASD) for females of 0.0156 (1.56%) and the assumed unbiased value for males of 0.0464. Method (2) became unreliable and was omitted. The new values are listed as follows as (method) P(ASD):(1a) 0.094, (1b) 0.074, (1c) 0.060, (5) 0.060, (6) 0.062.

The matrix method (5) was adjusted because with the reported estimated male and female ASD prevalence increase, the revised approximate range of true female P(ASD) became 0.055–0.065 (Figure 13). The new method (5) median value of P(AN|ASD) became 0.083 (range 0.031–0.164), exactly matching the hazard ratio value (Figure 14).

The period of data collection for my study closely matched that of the consecutive US studies [14,22]. Adding the five changed values of P(ASD) with the new data increased the total estimates to 13.

#### 2.4.2. A New Hazard Ratio for AN and ASDs

A recent study [23] looked at the sex differences in mental health problems in autistic young adults from 16 through 24 years from 2001 to 2013. There were 7129 ASD females and 5092 AN females in the database. In Table A1 of the appendix, the Swedish Register of Cox regression estimates of psychiatric diagnoses provides a value of the hazard ratio of AN in autistic females compared to AN in non-autistic females, i.e., P(AN|ASD)/P(AN|not ASD) of 3.92. This is equivalent to the Danish study and if we use the same arguments, we can use the equivalent current value for P(ASD|AN) of 0.25 with the odds form of Bayes’ theorem and derive a P(ASD) of 0.078. The key to the validity of the single value of the hazard ratio appears to be that the time frame extends through the majority of diagnoses of both conditions. The value of 0.078 is within the range of already derived values of P(ASD) and increases the total to 14.

#### 2.4.3. Summary of P(ASD) Findings

Despite the changes, the overall median value of P(ASD) has only moved from 0.0595 to 0.060, so there has not been any significant change in P(ASD) estimation with the new information. The final values of P(ASD) (Figure 13) give:Number: 14, Median: 0.060, First quartile: 0.059, Third quartile: 0.074, Range: 0.048–0.094.

Four methods were used to calculate the median value of P(ASD) as 0.060.

1(c). MFOR and bias calculations from my database and updated male ASD prevalence [1,22].

Hazard ratio and P(ASD|AN) [5,6,7,16].P(ASD) calculations from my study and AN data [1,5,6,7,13].

These three did have some overlapping independent variables.

3ASD and BPD [19,20,21]. The variables were quite independent of any of the methods above. 

The methods used in 3., 4. and 5. were versions of Bayes’ theorem.

The overall process has been to start with a Bayesian prior P(ASD) centred on 0.055, a value at the conservative end of the range found in the study determining the biases against female ascertainment [1]. Different versions of Bayes’ theorem were employed to update the variable, using multiple sources of information, to a median value of 0.060 with an interquartile range of 0.059 to 0.074. 

#### 2.4.4. Further Evidence for a Higher Female Prevalence

Defining ASD by DSM-5 we found a Male Female Odds Ratio (MFOR) of 3:4 [1]. The diagnosis by DSM-5 requires there to be a clinical problem. We hypothesized that, due to the large number of genes contributing to ASD, the most parsimonious MFOR value [24] for the genetic input to the diagnosis is likely to be 1:1. The major ongoing clinical problem in ASD is poor reciprocal communication and the greater socialization expectations and cultural pressure on girls [25,26,27,28] would tip the diagnostic balance to females as development proceeds. A recent report [29] followed a group of younger siblings of autistic probands from age 6 months to 5 years. By accounting for test bias against females over the observation period, they identified a group that had a high degree of ASD features and had a 1:1 gender ratio. My cross-sectional study [1] looked at all diagnosed siblings but in the age group up to 5 years, they were virtually all younger siblings. Recognition bias prior to diagnosis was assumed to be minimal because the carers knew what to look for due to their prior experience with ASD probands, and therefore, they would present them for an assessment of possible ASD. The diagnostic method was clinical (DSM-5), paying particular attention to the nuances of early female presentation [30,31,32] and 49% had prior testing, normally by allied health professionals (AHPs) (Table 7), using the Autism Diagnostic Observation Schedule version 2 (ADOS 2) and/or the Autism Diagnostic Interview-Revised (ADI-R). Gendered differences in the early presentation have been hard to tease out [27] probably because early diagnosed girls have behaved like the boys. Focusing on ASD siblings either longitudinally [29] or in the cross-section with informed carer history [1] appears to be the key to finding a more likely MFOR. My results are shown in Table 7 where the MFOR 95% confidence interval includes 1. 

While the study designs are different, the populations are essentially the same, as is the aim to establish the true female prevalence of ASD. If we take the view that the 1:1 result may reflect the true MFOR for the very young then the result from their study and the latest US result [22] which gives a value for male P(ASD) of 0.0464 implies a female P(ASD) of 0.0464, which meets the lower end of the range derived from the hazard ratio and P(ASD|AN) shown in Table 5. It nearly meets the lower range of the six methods of this paper of 0.048, and all of these six methods include children aged 1–18. Using the 1:1 ratio we would expect a result at the lower end of the range reflecting the true proportion that should be found up to 72 months. As more cases are found in any cohort after that time the prevalence would increase, by my estimate to about 6%. This would be due not only to the natural accumulation of cases diagnosed as older children and adults over time but also to the continuing social stresses converting female autistic spectrum conditions to autistic spectrum disorders. 

All these results are complementary and are converging on a much higher value of P(ASD) than current estimates.

### 2.5. Support for the Contention ASD Is Harder to Diagnose than AN

The Danish study also provides a hazard ratio for female ASD of 12.82 compared to 5.22 for female AN. We can then reverse the 2 × 2 table with ASD as a “test” for AN. P(AN) and P(ASD|AN) remain constant because they can be reliably measured. P(ASD) will be falsely low due to ASD bias relative to AN as a flag. If ASD is an inferior test for AN for this reason we would expect a falsely low P(ASD) and a falsely high P(AN|ASD) to result. The 2 × 2 table is shown in Figure 15. P(ASD) is 0.024 compared to 0.060 and P(AN|ASD) is 0.21 compared to 0.083 confirming the hypothesis. If the table had been reversed with the correct values on the right-hand side (b = 0.055 and d = 0.945) the LR would be 4.45 and not 12.82.

### 2.6. Incomplete but Suggestive Information of the Value of P(AN|ASD)

One publication [33] gave information on P(AN|ASD), but was not precise enough in male/female data discrimination to give a single value. The number of AN females was also a small possible range of 3–5 individuals but the range of female P(AN|ASD) can be calculated as 11.1–18.5%, which matches the top third of the range in Section 2.1. (Figure 16).

A second report [19] listed 4/40 ASD females as having an eating disorder without specifying the type. This would give a maximum value of P(AN|ASD) of 10%. Despite the small case numbers this value places it in the first half of the range in Section 2.1 and is consistent with the small number of estimates of P(AN|ASD). 

A third study [34] looked at the proportion of children with ADHD and/or ASD aged 9–12 years who had evidence of an eating disorder. They did not define the scope of eating disorders and the age group was at the lower limit for AN but noted it was indicative of future problems. The girls with ASD and ADHD had the highest rate of 9.7%.

### 2.7. An Example of the Clinical Value of the Methodology

It has been known for some time that ASD is associated with joint hypermobility and associated pain. At one extreme is Ehlers–Danlos Syndrome (EDS), predominantly hypermobile EDS (EDS3), but there now appears to be a continuum termed hypermobility spectrum disorder (HSD) [35,36] and it presents more commonly in females than males. Those who present with these features do so to physiotherapists and rheumatologists, and while these clinicians may be aware of the association, they probably will not be aware of the nuances of female ASD, in particular, the camouflaging behavior. We thus have a parallel with ASD and AN, where the HSD presents as a flag for ASD. In this situation, however, the state of the conditional probabilities is reversed. There is no reliable measure of P(ASD|HSD) in females and this value is critically important because if it is common then ASD must be diligently sought. There is a study in adult females [37] which gives values for a clinically significant subset of HSD, symptomatic generalized joint hypermobility (S-GJH). There are values for adult women of the probability of S-GJH in ASD, P(S-GJH|ASD) of 0.261, and the probability of S-GJH in a matched control group with ASD excluded, P(S-GJH|not ASD), of 0.070. With this information and a P(ASD) of 0.060, we can construct a 2 × 2 table (Figure A2 in appendix D) and derive a P(ASD|S-GJH) of 0.183. This suggests about 18% of females presenting to rheumatology and physiotherapy with S-GJH probably have ASD and up to 90% of them will be masking it [1]. This is potentially a serious problem [35] because HSD overall is not rare. The calculated female population prevalence of the subset S-GJH from the 2 × 2 table was 8.6%. The converse clinical situation is also vexing because 26% of females with S-GJH in the ASD population may have their joint problems ascribed to ASD alone [35].

To complete the comorbid circle there is good evidence [38,39] that there is a comorbid relation between eating disorders, including AN, and HSD. These genetic overlaps will be discussed in Section 3.4.

## 3. Discussion

### 3.1. The General Value of Bayes’ Theorem

The general method of using Bayes’ theorem to find the true prevalence of a cryptic condition is simple in principle. Find an associated condition which is more visible as a flag and easier to define and count. The difference in reliability of AN and ASD as a flag is shown by the difference between Figure 10 and Figure 15. Find the bidirectional conditional probabilities of the association and the prevalence of the flag condition and do the simple arithmetic.

Unfortunately, most mental health conditions do not have precise definitions, genuinely overlap and have varying diagnostic interpretations. The estimates often have wide ranges and do not report enough gender-specific results. My point estimates for calculations have used mid-range reported values in the hope that the data is sufficiently consistent and comparable to produce approximately normal ranges. A specific problem for female ASD is the dearth of available data for females. However, the results we have obtained for P(ASD), using different methods and multiple data sources, do converge on a value of 6.0% which is very different from current estimates and does have very real clinical significance. 

### 3.2. Clinical Value of the Relation between AN and ASD

The results show that there is a definite increase in the prevalence of AN in the population of women with ASD to about 8.3%, though it is a smaller proportion than the prevalence of ASD in the population of women with AN of about 25%. Since the occurrence of both in families was first reported by Gillberg in 1983 [40], the relationship between ASD and AN has become widely recognized by clinicians. Due to the cryptic female ASD phenotype, AN has become an important red flag for unrecognized ASD. ASD may have already been diagnosed, but for the majority of females, ASD has probably not been found by the time the peak rate of AN diagnosis approaches. ASD is comorbid with multiple eating disorders [41,42] and it is important not to lose track of the possibility of ASD in the noise. There is a potential problem if ASD in AN is not actually ASD but a manifestation of starvation in AN. There is now considerable evidence that ASD is truly present because the factors causing the disordered eating are different [12,13,43,44,45]. The therapeutic approach has to be different to allow for autism [45,46,47,48,49,50]. In the short term, compared to AN alone, there are longer and more frequent admissions and more severe general psychiatric comorbidity and need for antipsychotic medication [51]. Management is more difficult because therapists are often inexperienced in the different communication styles required to deal with ASD [52] and carers have more difficulty with aftercare because of a resulting lack of support [48]. In long-term follow ups, there are more mental health, psychosexual and socioeconomic problems. These are more related to ASD as the eating disorder has generally resolved [12]. It thus becomes very important to recognize ASD in AN or a satisfactory psychosocial outcome will be very difficult to achieve, in particular, in the long term.

### 3.3. The Significance of ASD Prevalence

The most important result of this paper is the probability that the prevalence of ASD in women is much higher than previously thought. Pediatric clinicians know we are missing girls before they pass out of our care, but we may be missing a frightening number of girls. If we have an accurate estimate of both P(ASD) and the prevalence of the comorbidity we only need one of the conditional probabilities to establish the complete quantitative relationship. This was demonstrated by the first outcome of this paper, establishing the unknown P(AN|ASD). The only range of unbiased values for P(ASD) was from my first paper, but the current study has led to two further methods of establishing P(ASD), for a total of 14 estimates by six methods. These strongly suggest a prevalence of about 6%. Unrecognised ASD has a large number of psychiatric comorbidities [53,54], and unless the upstream factor of ASD is diagnosed, it is likely that anorexia nervosa is only one of many mental health conditions in which this methodology could be explored, with consequential improvements in diagnosis and management of complex conditions.

### 3.4. The Structure and Significance of Comorbidity

Medical nosology, including psychiatry, has a history of being categorical. This makes general sense in terms of simplifying diagnosis, management and prognosis, but there is increasing evidence that most mental health conditions are strongly polygenic, and that the genes tend to be pleiotropic, i.e., a single gene affects multiple phenotypic traits, to the extent that many, if not most, conditions overlap [55,56,57]. Generalized joint hypermobility, including hypermobile EDS, appears to be polygenic [36] and so the genetic pleiotropy appears to extend beyond mental health diagnoses. 

The diagnostic field of mental health is multidimensional and has a continuous rather than a categorical landscape. There are interactions within and between different levels. The hierarchy of complexity consists of genes, genetic modifiers such as copy number variations, single nucleotide polymorphisms and epigenetic alterations [58], the physical, chemical and biological milieu at molecular and cellular levels, gender, the effect of neurodevelopmental adaptation, life events and sociocultural stressors. Our genetic software is a recipe, not a blueprint. No two outcomes are exactly the same. Random events during neurodevelopment produce different phenotypes, as demonstrated in identical twin studies [59].

The autistic spectrum is not one-dimensional like the electromagnetic spectrum. It is multidimensional starting with a complex of ASD phenotypes which then combine with comorbid conditions, such as ADHD, mood disorders, borderline personality disorder, psychoses and eating disorders. These then interact in different ways over time [60], as outlined above. The phenotypic outcomes overlap resulting in converging overt behaviors which can be very challenging to untangle in terms of precise condition formulation, in particular for high-functioning adult women [61]. An individual will have a lifetime risk due to the underlying genetic makeup and defining the situation will change in both character and severity over the life span. Whether the phenotype at any given time is a particular set of disorders and/or conditions will depend on the balance of environmental stressors and effective therapies.

### 3.5. Generalizing the Importance of the Prevalence of Female ASD

Female ASD is then cryptic, common and comorbid with mental health conditions (MHC) in general and other relevant disorders, such as S-GJH. While the proportion of ASD comorbidity will vary with condition, age and circumstance, it is useful to look at the overall picture to get a feel for the extent of the problem. The lifetime prevalence of MHC is about 30% for both sexes [62]. Literature values for the proportion of mental health conditions in autistic people, P(MHC|ASD), are generally high [63] and we will use 80% as a likely reasonable estimate. Then, with female P(ASD) of 6%, by Bayes’ theorem, the proportion of MHC with ASD is:
P(ASD|MHC) = P(MHC|ASD) × P(ASD)/P(MHC)= 0.8 × 0.06/0.3= 0.16

On average 16%, or one in six women who have a mental health condition, have comorbid ASD. Consider the conditions we have looked at in this paper (Table 8).

Half of all individuals with a mental health disorder have their onset by 18 years, 62.5% by age 25 [64]. This is a problem of and for the young in particular, but it is crucial that clinicians consider ASD in any mental health condition at any age, especially if management is failing.

How this information translates into an overall model is currently a matter for debate [65]. How, then, is the clinician to manage in practice?

### 3.6. Clinical Implications

#### 3.6.1. Diagnosis and Treatment

Since female ASD is cryptic and common, it poses specific challenges for management. It is critical that the ASD signal is not missed in the phenotypic noise. If the criteria for ASD are met within the complex phenotype of an individual patient, the diagnosis should be made. Even if the features are “subdiagnostic”, they should be considered in tailoring management since therapeutic success is going to be more difficult than if the features are absent. The complexity and fluidity of an individual’s phenotype mean the therapist should show considerable humility in making a diagnostic formulation, which then needs to be kept under review. It also means that management needs to be pragmatic. Management of these overlapping conditions requires self-knowledge by the patient, a supportive environment and appropriate interventions, often including medication [66,67,68,69]. The increasing knowledge of ASD at the molecular level over time will improve the accuracy of medication targeting [70,71]; meanwhile, clinicians need to be flexible in finding what works. 

#### 3.6.2. Diagnostic Controversy

A paradigm of pediatric training is that the medical history, particularly that of the mother, is discounted at your peril. The parents of children with disabilities are routinely disbelieved, in particular in conditions which have no overt physical signs, such as high-functioning autism [72]. This is particularly so for girls, where their skills at camouflage and mimicry, everywhere but home [8], make corroborating evidence scanty. 

“The reward for trying hard to be normal was to be ignored because you were acting normal and I look at stories online of kids who were going off the rails and I think, I should have just burnt more cars.” [26].

Girls, for example, show evidence of camouflaging behavior at school in 88% of cases [1].

“I was unbearable with my mother, but at school I was perfect.” [26].

The maternal history is even conflated with fabricated illness and can become a child-protection issue. This is causing serious distress in the families and the neurodivergent community [72]. The diagnostic process can be very stressful for all concerned. The diagnostician has to juggle the conflicting interests of the diagnostic tools, the patient and family narratives and predicting the positive and negative consequences of diagnosis for patients and families [73]. 

In a gray area, you need to establish a default position and have a plan to manage it. However, you need to be clear where the actual gray area is, and there are two.

In the absence of corroboration, how reliable do you believe a convincing history to be?Is the overall picture including the history equivocal? 

My own practice in area one is to go with my pediatric training. The history will include a careful family history due to the high genetic content for ASD and its comorbidities. It is unfortunate that social and mental health issues in the family often count against the perceived veracity of the history when in Bayesian terms they actually reinforce the probability of ASD due to the mental health comorbidities. If in the future, the diagnosis is not sustained, no real harm is done since either the “disorder” is no longer present or a more viable mental health diagnostic formulation will be being treated. However, be mindful that ASD may have been successfully converted to ASC by prior intervention.

In area two, a second opinion and testing by a skilled psychologist will be helpful. If things remain equivocal, offer ongoing surveillance. If a crisis point is reached, then reassess.

While testing may be useful as an adjunct to diagnosis, a recent report implies that, despite ADOS testing being regarded as a necessary part of a “gold standard” diagnosis, it is neither necessary nor sufficient in children under 6 years of age [74]. At all times uncertainty mandates ongoing monitoring.

#### 3.6.3. Autistic Spectrum Disorder and Autistic Spectrum Condition

The distinction between ASD and ASC is confusing. They may be thought synonymous but used to convey different philosophies of autistic neurodiversity [25]. A more useful distinction would inform a rational clinical approach. We would define an ASC as a set of neurodiverse behaviors due to a person’s biological software (genetic and epigenetic) which becomes ASD when enough stress from genes to culture causes a transition to definable psychopathology. Then, ASD becomes a clinically meaningful subset of ASC requiring a different approach, which we will now discuss.

The study by Burrows et al. [29] implies a 1:1 ASD MFOR, largely from a genetic source and my data in Section 2.4.2 supports this. However, the MFOR up to age 18 in my study [1] was 0.75 (3:4), whereas at age 7 it was 0.85. The studies together support the conversion of ASC to ASD in females by environmental stress over time, as described in Section 3.2. The conversion of potential to actuality has been described [59] as a developmental consequence of an allostatic load of earlier-occurring liabilities, indexed by early behavioral phenotypes, in varying permutations and combinations. The clinical threshold can be viewed as a tipping point. This highlights the importance of early diagnosis to prevent or minimize the stresses. It also raises the issue of what then we should be diagnosing. Specifically, should the aim be to identify ASC and intervene? If so with what goal? My view is that ASC is not a disability and the aim should not be to try to make the person neurotypical. Even after removing the high outlying P(ASD) estimate of 9.4% (Figure 13), the values are skewed above the median. If we accept ASD as a subset of ASC, this suggests autistic neurodiversity is common. If ASC is identified, the aim of any intervention should be to prevent the deterioration to ASD. If ASD is diagnosed the aim of therapy should be to convert ASD to ASC as the logical endpoint [1]. Disorder is defined more by comorbidities than the features of neurodiversity. The focus of any therapy should be on disorder rather than autism [75].

#### 3.6.4. The “Aspie” Experience

A long-term strategy guided by patience is essential. A diagnostic formulation and a way forward need to be negotiated with a communication style that the autistic person is comfortable with, including, for example, close-ended questions, a support person if needed and adequate time to respond, including written responses. The effect of comorbid anxiety must never be underestimated. A useful analogy may be trying to get by in a foreign language and the languages of “neurotypical” and “aspie” are often barely mutually intelligible.

Consider the problem from the perspective of the autistic person. Obviously, the better you can handle a foreign language the less stressed you will be. If you can only just cope it is exhausting work. Body language and cultural norms are different. Reading is easier than conversation because you have the time to process it. You hope the foreigner does not think you are intellectually impaired or mentally ill. It might be safer to shut up (if I feel in control) or shut down (if I do not feel in control) and say nothing. Where the analogy fails, is the foreigner will usually allow for the fact that you are not a local, but with ASD you are desperately pretending to be a local and the other party is often unsympathetic, “And the shouting really hurts my brain!”.

We all camouflage depending on the social situation but imagine having to do it in a foreign language and culture you do not really understand when interacting in your own society every day.

#### 3.6.5. How to Succeed

Intrafamilial clashes are common since ASD is highly genetic. The adults need to understand this and adapt. The therapists need to listen (Figure 17), not only to the patient and family but to each other. If the therapists and the key family members are not on the same page then management will fail. For school-age children, the teacher is a key therapist. Achieving harmony is not easy but if it is achieved a satisfactory outcome is highly likely.

In particular, a top-down approach does not work in autism therapy. This applies both to the patient, and if appropriate, to the family, who have to implement the plan. 

The wise clinician, personifying the Art of management, will adopt the following approach, adapted from Chapter 17 of the Tao Te Ching, attributed to Lao Tzu [76]:

The best therapist values her words and uses them sparingly.

When she has accomplished her task the patient says:

“Amazing! I did it all by myself!”.

## 4. Strengths and Weaknesses

One clinician can drill down deeply into qualitative aspects and quantitative variables of individual patients. These may include gender, referral patterns and details of where they fit in families, especially blended families. Comparative studies can be very finely focused. Due to the personal relationship between the clinician and the patient/family, it is easy to get a study population which reflects the local drainage population. My experience has been that the families are very aware of the difficulties that autistic females have to deal with, and the participation rate where needed has been 100%. 

However, there is a greater risk of systematic errors, such as diagnostic bias or local patient variation not reflecting a larger population. Absolute prevalence studies cannot be done and smaller numbers make time-based comparisons inaccurate. The weaknesses can be overcome if there are sufficient published studies but in mental health, they are often sparse in terms of gender specification and age range, and the results are widely variable due to the imprecision of diagnostic specificity and comorbid overlap.

## 5. Future Directions

Be gender specific in epidemiological and clinical reporting. Try to think of more ways to establish female ASD variables: epidemiology, clinical features, diagnosis and management. If the greater true prevalence of female ASD is confirmed, take it much more into consideration in clinical management. Take advantage of comorbid variables to use Bayes’ theorem to fill multiple knowledge gaps. It appears that mental health conditions are continuous and not categorical, and this is going to be a challenging task. While the method is important for overlapping mental health conditions, it will also apply to any comorbid condition. For anorexia nervosa, rigorously search for comorbid ASD and improve management techniques in the short and long term to improve prognosis.

## 6. Conclusions

There appears to be a prevalence of about 8–9% of anorexia nervosa in female ASD by about the fourth decade.

There are at least four factors contributing to it not having been quantified:

Female ASD is cryptic.Adult female ASD services are fragmentary.Peak AN time of diagnosis coincides with the adolescent to adult transition.The proportion of AN in ASD is elevated above AN prevalence but not dramatically so.

There appears to be a prevalence of about 6% of female ASD by about the third decade.

The data for both conditions is still sparse, and the lifetime prevalences are likely to be even higher as the diagnosis of adults improves.

Autistic neurodiversity in females is common. The prevalence is unknown but is likely to be significantly greater than 6%.

Female ASD is still poorly recognized and it appears to be a common comorbidity in mental illness with one in six females only partially diagnosed at best. This would be a common causal factor of treatment failure in these illnesses. 

Most females camouflage their ASD. Our best diagnostic guide for girls is the mother’s history. Our best diagnostic guide for women is to think of it. The key skill in diagnosis and therapy is listening.

## Figures and Tables

**Figure 1 children-10-00272-f001:**
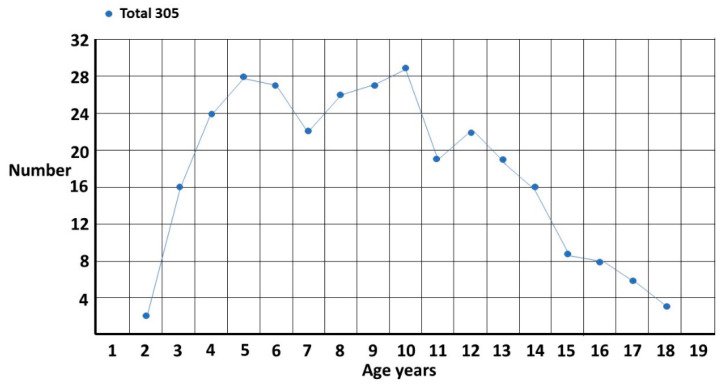
Female singletons diagnosed per year of age.

**Figure 2 children-10-00272-f002:**
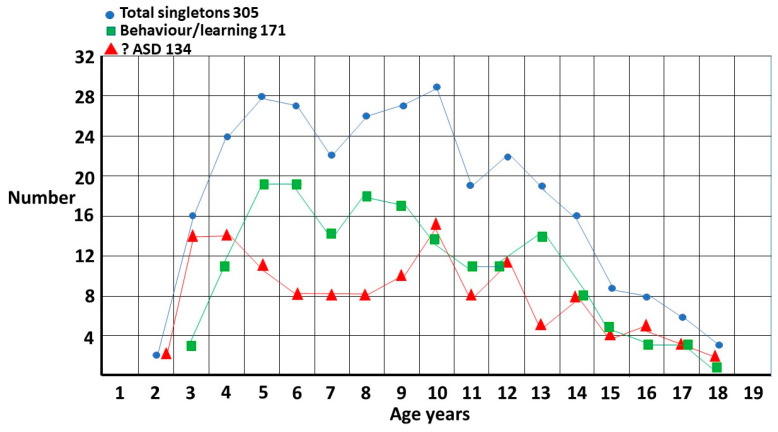
Singleton distribution by referral question. ?ASD and behaviour/learning now referenced in text twice, before and after figure.

**Figure 3 children-10-00272-f003:**
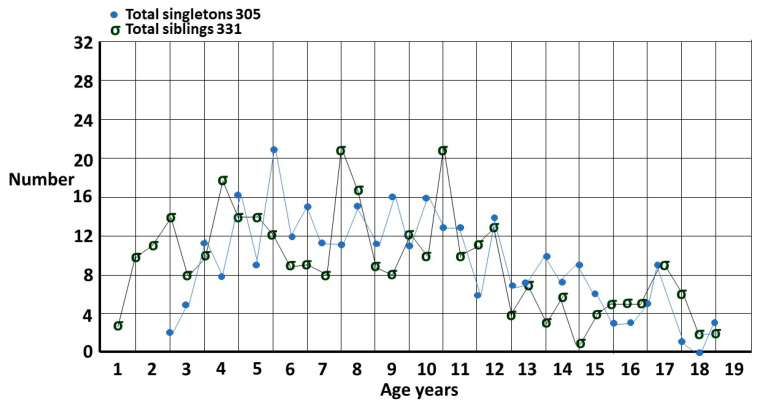
Singletons and siblings for each 6 months of age.

**Figure 4 children-10-00272-f004:**
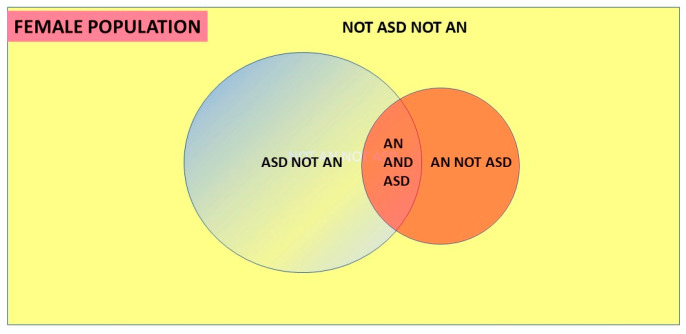
Venn diagram of Bayes’ theorem.

**Figure 5 children-10-00272-f005:**
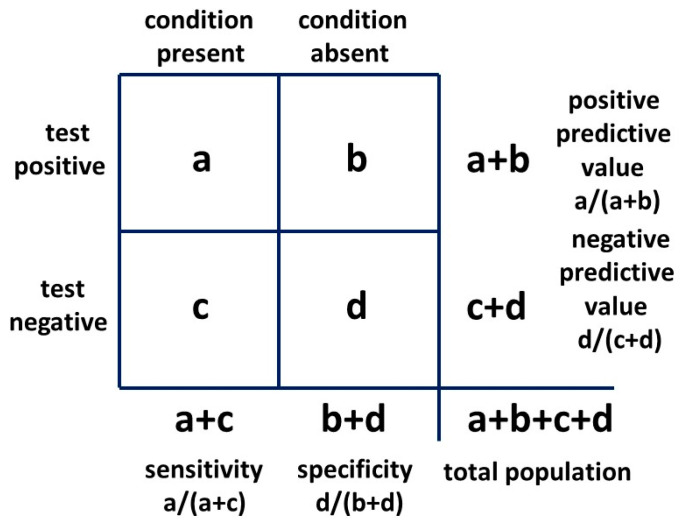
Bayesian 2 × 2 table.

**Figure 6 children-10-00272-f006:**
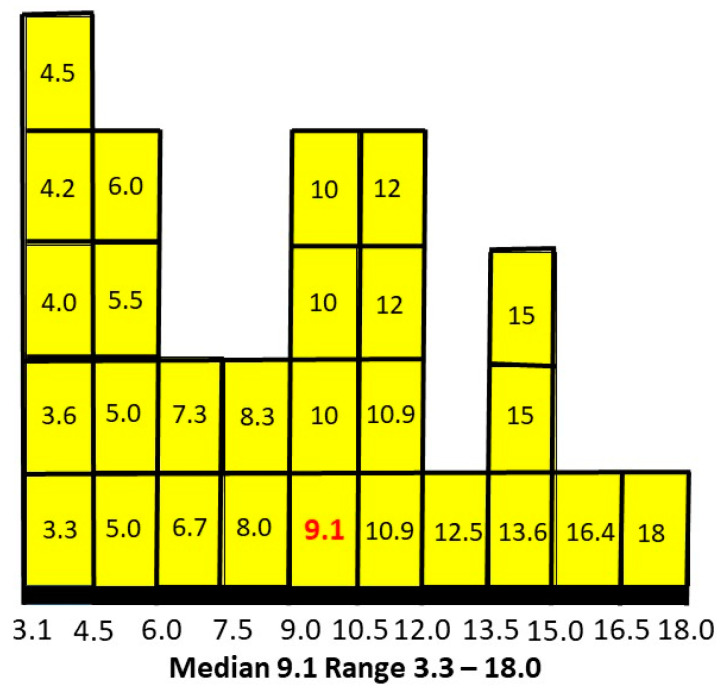
P(AN|ASD) distributed as percentages.

**Figure 7 children-10-00272-f007:**
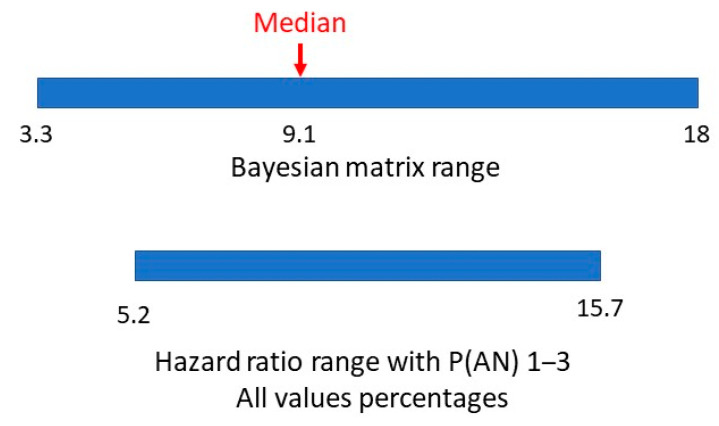
Bayesian matrix and hazard ratio ranges of P(AN|ASD).

**Figure 8 children-10-00272-f008:**
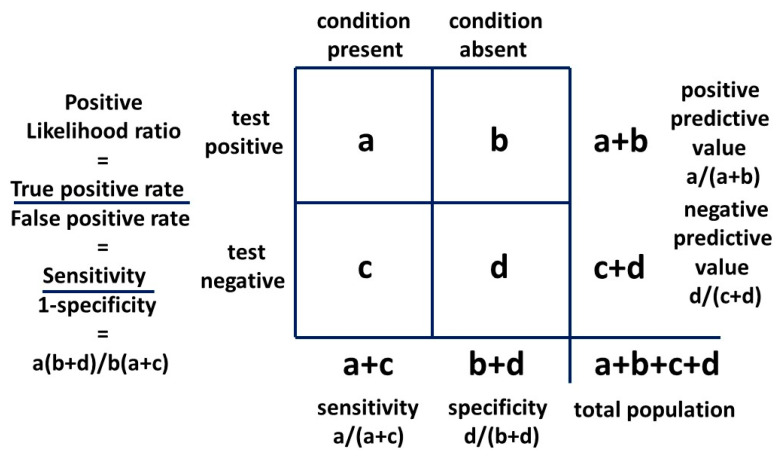
Bayesian 2 × 2 table.

**Figure 9 children-10-00272-f009:**
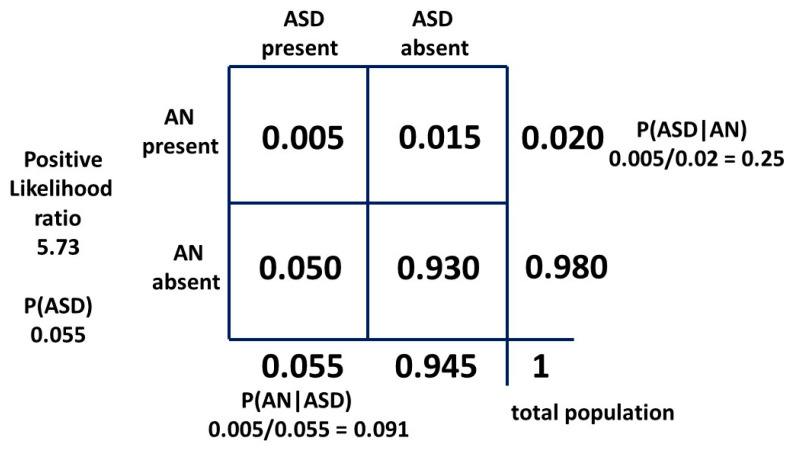
Bayesian 2 × 2 table for female model one as probabilities.

**Figure 10 children-10-00272-f010:**
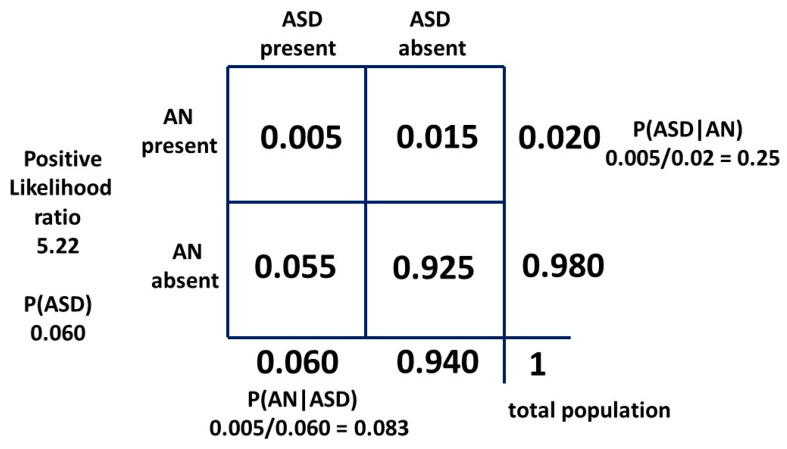
Bayesian 2 × 2 table for female population model two as probabilities.

**Figure 11 children-10-00272-f011:**
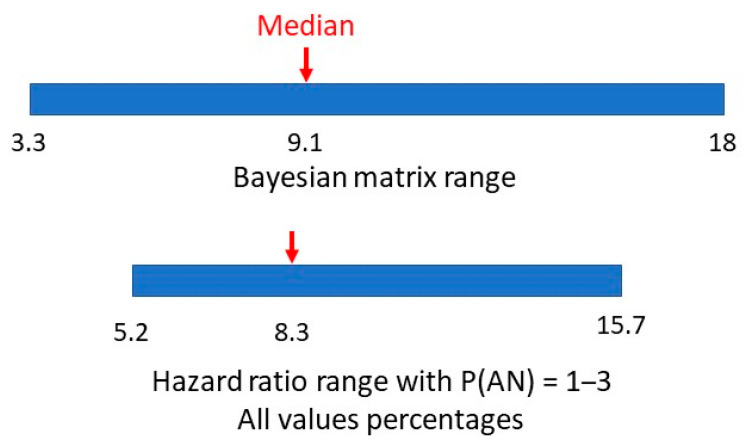
Bayesian matrix median and hazard ratio value and ranges of P(AN|ASD).

**Figure 12 children-10-00272-f012:**
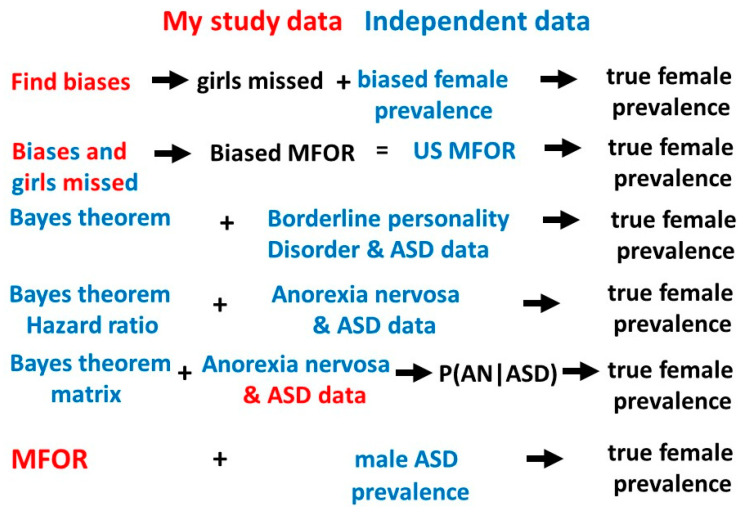
Pathways to the true female ASD prevalence.

**Figure 13 children-10-00272-f013:**
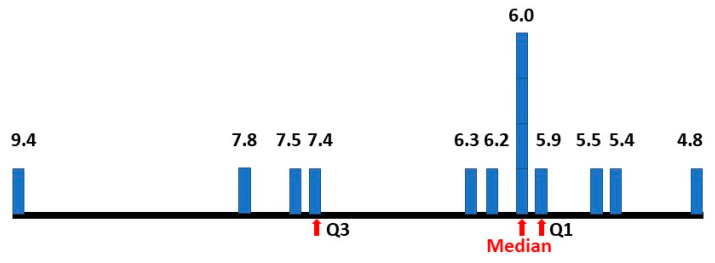
Percentage prevalence estimates of female ASD.

**Figure 14 children-10-00272-f014:**
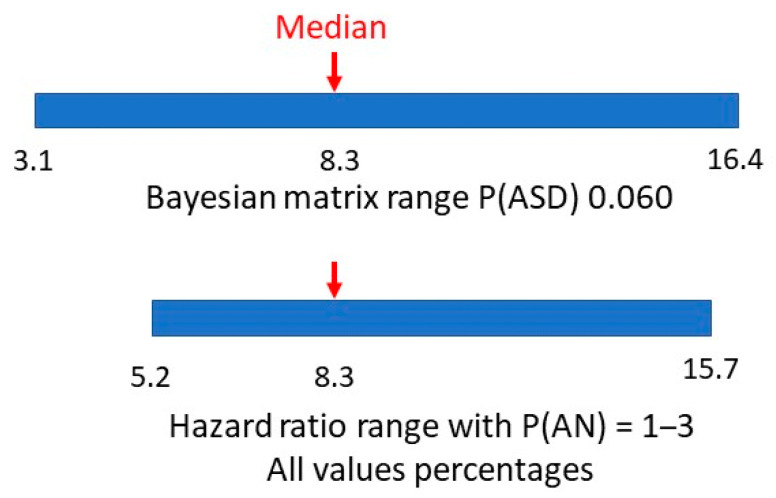
P(AN|ASD) revised Bayes’ matrix range.

**Figure 15 children-10-00272-f015:**
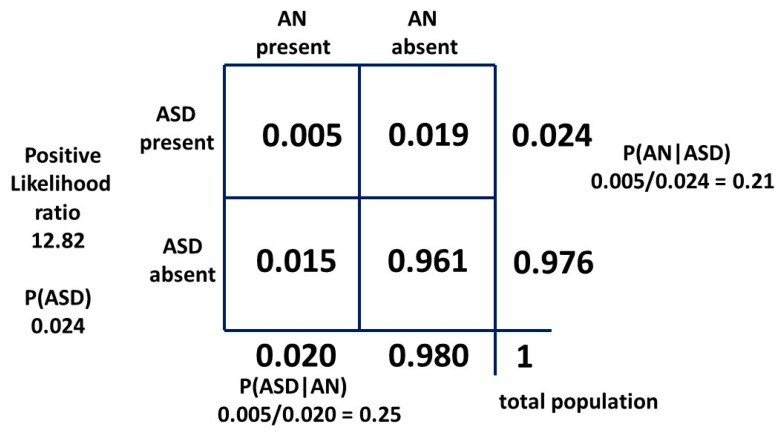
Bayesian 2 × 2 table with ASD as a test for AN.

**Figure 16 children-10-00272-f016:**
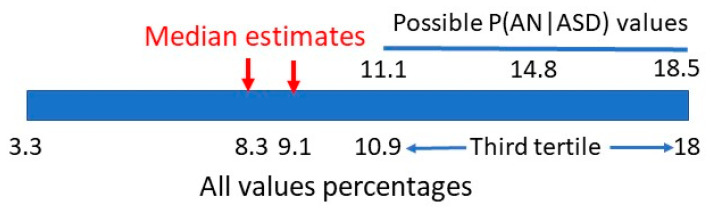
Bayesian matrix range of P(AN|ASD).

**Figure 17 children-10-00272-f017:**
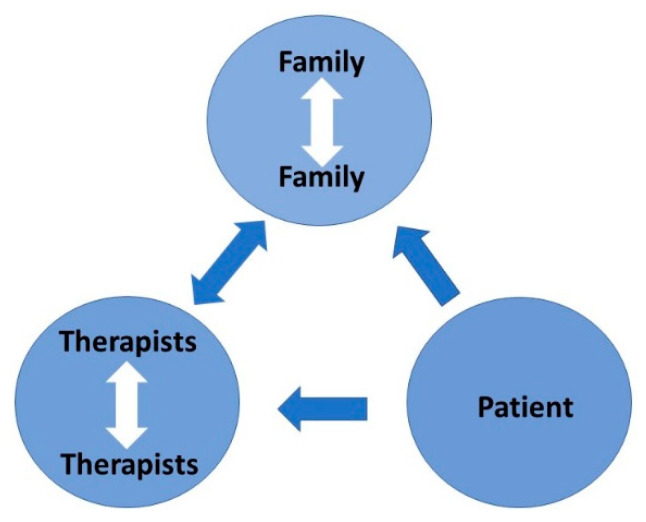
Therapeutic dynamic: the key is to listen.

**Table 1 children-10-00272-t001:** Median times to diagnosis for each female category.

Patient Category	Median to 96 Months	Number	Median to 18 Years	Number	96 Months Proportion %
Total singletons	5y 7 m	121	9y 2 m	305	39.7
Singletons: ?ASD	4y 10 m	55	9y 4 m	134	41.0
Singletons: behavior	6y 0 m	66	9y 0 m	171	38.6
Siblings	4y 9 m	161	8y 1 m	331	48.6
All patients	5y 4 m	282	8y 4 m	636	44.3

**Table 2 children-10-00272-t002:** Probability ranges for independent variables.

Variable	Studies	Subjects	Low	Mid	High	References
P(ASD|AN)	8	327	0.20	0.25	0.30	[2,6,7]
P(AN)	>125	>700,000	0.01	0.02	0.03	See Section 2.1
P(ASD)	1	1711 *	0.050	0.055	0.060	See Section 2.1

* Includes males for calculations.

**Table 3 children-10-00272-t003:** Values of P(AN|ASD) for the ranges of the variables by Bayes’ Theorem.

P(ASD|AN) × P(AN)	P(ASD)		
	0.050	0.055	0.060
0.20 × 0.01 = 0.002	0.040	0.036	0.033
0.20 × 0.02 = 0.004	0.080	0.073	0.067
0.20 × 0.03 = 0.006	0.120	0.109	0.100
0.25 × 0.01 = 0.0025	0.050	0.045	0.042
0.25 × 0.02 = 0.005	0.100	0.091	0.083
0.25 × 0.03 = 0.0075	0.150	0.136	0.124
0.30 × 0.01 = 0.003	0.060	0.055	0.050
0.30 × 0.02 = 0.006	0.120	0.109	0.100
0.30 × 0.03 = 0.009	0.180	0.164	0.150

where: P(AN|ASD) = P(ASD|AN) × P(AN)/P(ASD). Red is to highlight the median.

**Table 4 children-10-00272-t004:** P(AN|ASD) and P(ASD) estimates.

Study	P(AN|ASD)	P(ASD)
1a. Data from proband biases and US biased ♀ prevalence	0.067	0.075
2. Study estimate from US MFOR	0.080	0.063
3. Bayes’ and Borderline Personality Disorder	0.083	0.060
4. Bayes’ and AN hazard ratio	0.083	0.060
1b. Data from singleton biases and US-biased ♀ prevalence	0.085	0.059
5. Bayes’ matrix and AN data	0.091	0.055
6. Study MFOR and US male prevalence	0.092	0.054
1c. All study ♀ and US biased ♀ prevalence	0.104	0.048
Median values	0.084	0.0595

P(AN|ASD) = 0.005/P(ASD) assuming that P(AN) is 0.02 and P(ASD|AN) is 0.25.

**Table 5 children-10-00272-t005:** P(ASD) for range of P(ASD|AN) with hazard ratio of 5.22.

P(ASD|AN)	P(ASD)
0.20	0.046
0.25	0.060
0.30	0.076

**Table 6 children-10-00272-t006:** P(ASD) for 95% range of hazard ratio with P(ASD|AN) of 0.25.

Hazard Ratio	P(ASD)
6.64	0.048
5.22	0.060
4.11	0.075

**Table 7 children-10-00272-t007:** Variables for sibling diagnosis by age.

Age in Months	<24	24–35	36–47	48–59	60–71	Total
Male number	11	13	23	23	22	92
Female number	13	19	19	30	20	101
MFOR	0.802	0.649	1.147	0.727	1.043	0.863
% AHP help in diagnosis	69.6	86.7	51.4	44.2	42.3	49.2

MFOR: 95% confidence interval 0.660–1.166.

**Table 8 children-10-00272-t008:** Proportion with ASD in comorbid conditions.

Comorbidity	% with ASD	References
Borderline personality disorder	14.6	[1,20]
Anorexia nervosa	20–30	[2,6,7]
Symptomatic generalized joint hypermobility	18.3	Section 2.7, [37]
Any mental health condition	16	[62,63]

## Data Availability

Data for the epidemiology section assisting patient categorisation eg gender, chronological age and age at diagnosis has not been deidentified and is not presented. It would be unique to any study and therefore is not transferable.

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
