# Peer review of "Finding the Proportion of Females with Autistic Spectrum Disorder Who Develop Anorexia Nervosa, the True Prevalence of Female ASD and Its Clinical Significance"

_children, 2023, doi:10.3390/children10020272_

Round 1

Reviewer 1 Report

Overall, this paper needs to find its focus and then can be significantly shortened (potentially to a third of its length). Starting with the title and abstract, it is difficult to understand the point of the study. I also caution the general use of the term "mental health" in this paper, as it wasn't clearly defined and it was not clear if they were calling ASD a mental health condition or using AN as a proxy for all mental health problems. Lines 43-47 are the clearest point of this paper and this should be highlighted much earlier, and this point should be returned to throughout the paper.  

Significant amounts of the introduction should be cut out, as it was previously published by the author and should not be repeated at length.

The methods section should also be shortened, as much was spent on explaining general methodology rather than only the amount required for the actual work presented. 

Reviewer 2 Report

This is an elegant study on the proportion of females with ASD who develop anorexia nervosa (AN). Generally, the study is well designed and written. I have only 2 general remarks:

1.  The Introduction should include some general information on ASD and AN for the readers unfamliliar with the subject

2. Does the Author have any observations on the relationship between the outcome of AN treatment in girls with concomitant ASD? It has been demonstrated that AN is characterized by high relapse rate and the treatment outcome is associated with initial BMI and hospitalization time. Probably it should be mentioned as at least further study perspective.

Reviewer 3 Report

Congratulations on such a great scientific work and presentation of research from your own clinical practice and other epidemiological research. This article has contributed significantly to the current knowledge, showing comorbidity and prevalence of autistic spectrum disorder and anorexia nervosa in girls. The introduction is clear and exhaustive, the methods are described in details, the results and equations transparent, discussion and conclusions adequate to the content and important for the whole society, including patients and their family, clinicians and therapeutists. I have only three suggestions for minor changes:

1. Please report and comment on the effect size for the Chi-square test, e.g., Cramer's V (page 4, line 128).

2. Please add an explanation for all abbreviations, when first presented in the text (e.g., ICD, DSM, ADOS, ADI-R, AHP help in Dx, ADHD). In particular, it is unclear what "P" means in this pattern P(AN|ASD): prevalence, proportion, percentage, or probability?

3. It would be helpful if you would report the number of children in each study cited, including your own.

Round 2

Reviewer 1 Report

Thank you for your response regarding the concern there was repeated information from the previous manuscript – it is more clear now that you state in your introduction that it was your dataset and that you are building upon previous work.

Revised section of the Pre-amble was very nicely done and makes it much more clear the purpose of the paper. I would add a citation to the statement “AN is a serious eating disorder, predominantly in females, which is life-threatening if neglected and is relatively easy to detect.” And “…ASD is that it may be a common comorbidity of many mental health conditions.”

Table 2 – should be more clearly cited where these data are from (can be added in text when you state to refer to table 2), was it from citation 1? Or 5, 6, 7, 13?  This was the preceding text: “My selected ranges were of course educated guesses, but were based on current available data for AN lifetime prevalence [13], P(ASD|AN) [5,6,7] and prevalence of ASD. For the first two there are quite a lot of studies and the ranges can be established with reasonable confidence. This is not so for ASD prevalence and the range is based on assessments from my study of the proportion of females missed [1], where a range of possible P(ASD) prevalences can be calculated both from my data and independent data as explained in that study.” 

Fig 12: needs cleaning up and not in red and green given readers with color blindness will not be able to see it. It is not clear if the colors make an additional impact to the figure, and they detract from the information.

Thank you for adding such helpful information about the clinical impact of your findings in the discussion. I think that the first part of the discussion (3.1 – 3.4) is well done and flows nicely with the rest of the paper. However, 3.5  could be shortened (potentially is not needed, as 3.3 and 3.4 says a lot about the clinical complexity and impact and does it well). If the goal is to provide clinical tools for providers, then it would help to have less background information interfering with the author’s overall point. For example, in 2.5.2 Diagnostic controversy, the author could keep the first paragraph and last three paragraphs.

3.5.3 Comorbidities in school age girls – this section seems to come on abruptly – was this data previously reported (if so, please cite)? Or was this a random sample obtained in response to your findings reported previously? I would clarify where and why this data was performed, and if it was previously planned, I would recommend adding it to your methods. I would also clarify “school age girls (singletons and siblings) from my clinic” to “school age girls (singletons and siblings) with autism seen at my clinic” as readers don’t know if patients have to be previously diagnosis of ASD to be seen at your clinic (vs. some come to you for a rule out diagnosis). Overall, if this is brand new data, I think it would be better to just cite other studies about mental health comorbidities in ASD as this is just describing one clinic’s treatment of comorbidities (rather than actual description of common comorbidities, if that is what the aim of this section was). If you plan to keep this data here, I would still add other studies.

3.5.4 This section may do better in the first part of the discussion, OR could consider cutting much of it to highlight the actual clinical recommendations that are scattered throughout this section.

 “The move with age to a female preponderance rather suggests a male environmental protective effect due to the lesser sociocultural pressures on males. A woman is supposed to discuss feelings and relationships over coffee. Autistic women hate that. Camouflage as a protective mechanism for women either fails, or “succeeds” at significant psychological cost. A man on the other hand is allowed to grunt and play by himself in his shed with his power tools.” This is a VERY risky comment to make without citations. Although we all know there are gender norms and I am sure you are correct on some level, please cite studies about gender norms and their impact (it does not have to be on ASD). Or remove as I’m not sure if you have enough evidence to say this finding suggests only an environmental, sociocultural protective effect.

3.5.5 The “Aspie” experience – “Autistic people do not like being told what to do…” though this may be anecdotal, this is a broad statement that could be applied to any group. Please cite, or could consider just removing the first line. In this section, about halfway down the use of “you” starts and it is confusing if it is referring to people with ASD (which was previously referred to as ‘they’ or if the author is talking to the reader. I think citations are needed in this section in general.

3.5.6 – this is a great section. Fig.17 may need cleaned up – I know the point is to listen, but it looks messy (the arrows already make that point).

Future Directions – great points through and through. Can the author clarify “we are still missing ASD males as well?” I’m not sure what we are missing. Underdiagnosing?

Conclusions  - “3. Peak incidence coincides with adolescent to adult transition” – this isn’t a fair interpretation of the peak incidence at 15-19 – it largely onsets in 13-17, right in the middle of adolescence, but not diagnosed until these ages. Just would caution that interpretation. “4. The proportion of AN in ASD is elevated beyond background but not dramatically so.” Please clarify “beyond background” – not sure if that is a regional phrase.
